# Efficient Uzawa algorithms with projection strategies for geodynamic Stokes flow

Deok-Kyu Jang<sup>1</sup>, Kyeong-Min Lee<sup>2</sup>, Cedric Thieulot<sup>3</sup>, Whan-Hyuk Choi<sup>1,4</sup>, and Byung-Dal So<sup>2,5</sup>

**Correspondence:** Byung-Dal So (bdso@kangwon.ac.kr)

**Abstract.** Stokes equations are often difficult to handle in geodynamic modelling because they form a saddle-point system and involve strong variations in viscosity. Uzawa-type solvers are straightforward to implement, but their convergence may become slow if a suitable preconditioner is not used. Here, we introduce two adjustments that improve stability and efficiency. Residuals are evaluated in weak form, giving an effect similar to that of a mass-matrix preconditioner. We also add a projection step so that the velocity field remains nearly divergence-free. These updates made the solver converge faster and behave more stably than the standard Uzawa method. The modified approach was tested in several cases, including ABC flow, SolCx, mantle convection, block sinking, and compressible convection under the Anelastic Liquid Approximation.

# 1 Introduction

Geodynamic modelling of the mantle and lithosphere is concerned with processes such as subduction (Gerya, 2022; Li and Gurnis, 2023; Sime et al., 2024), continental rifting (Huismans et al., 2001; Brune et al., 2012; So and Yuen, 2015), mantle plume upwelling (Ismail-Zadeh et al., 2006; Negredo et al., 2022; Kim and So, 2020), and lithospheric buckling (Do et al., 2023; Xie et al., 2024). The mantle has high viscosity, and the related flows occur at low Reynolds number. As a result, inertial terms are negligible, and the Navier-Stokes equations reduce to the elliptic Stokes system. This formulation produces large and sparse saddle-point systems that are challenging to solve, especially when viscosity varies by several orders of magnitude and complex rheologies are present (Moresi and Gurnis, 1996; Zhong et al., 2000).

The spatial discretisation of the Stokes equations in geodynamics is frequently performed using finite element (FEM), finite difference (FDM) procedures, or finite volume (FVM) techniques. These algebraic systems retain the saddle-point structure of the velocity–pressure formulation. For this reason, direct solvers are not a realistic method for three-dimensional problems due to memory and computational expense. Due to this reason, iterative Krylov subspace approaches such as the conjugate gradient (CG) method for symmetric positive definite systems (Hestenes et al., 1952) and the generalized minimal residual method (GMRES) or its flexible variant (FGMRES) for nonsymmetric systems (Saad and Schultz, 1986; Saad, 2003) are often

<sup>&</sup>lt;sup>1</sup>KNU Research Institute of Mathematical Sciences, Kangwon National University, Chuncheon, Republic of Korea

<sup>&</sup>lt;sup>2</sup>Department of Interdisciplinary Program in Earth Environmental System Science & Engineering, Kangwon National University, Chuncheon, Republic of Korea

<sup>&</sup>lt;sup>3</sup>Department of Earth Sciences, Utrecht University, Princetonlaan 8A, The Netherlands

<sup>&</sup>lt;sup>4</sup>Department of Mathematics, Kangwon National University, Chuncheon, Republic of Korea

<sup>&</sup>lt;sup>5</sup>Department of Geophysics, Kangwon National University, Chuncheon, Republic of Korea

55

used for large-scale simulations. In the absence of suitable preconditioning, such methods progress slowly, especially when strong viscosity contrasts reduce the conditioning of the operator.

A common strategy is to eliminate the velocity variables and form a reduced Schur complement system for the pressure. The standard Uzawa algorithm (Arrow et al., 1958) applies Richardson-type iterations to this system and recomputes velocity at each step. The method is simple but converges slowly and is sensitive to the relaxation parameter, which limits its use in geodynamic applications. Later studies proposed variants with preconditioning and inexact solves, such as block preconditioners and mass-matrix approximations, to improve efficiency and stability (Cahouet and Chabard, 1988; Elman and Golub, 1994; Chen, 1998; Bai and Wang, 2008). Braess (Braess, 2001) presented explicit formulations of adaptive Uzawa and conjugate-directions Uzawa, showing their relation to gradient and conjugate gradient iterations. In geodynamics, Uzawa-type updates have been implemented in mantle convection codes such as CitcomS (Zhong et al., 2000), and scalability studies have evaluated their performance on large parallel systems (Gmeiner et al., 2016). More generally, Schur complement systems can be solved using Krylov methods combined with suitable preconditioners, and much work has focused on block preconditioners, multigrid methods, and algebraic strategies for Stokes flow (Silvester and Wathen, 1994; May and Moresi, 2008; Kronbichler et al., 2012). These approaches are now widely adopted in large-scale mantle convection modelling.

Modern geodynamic codes no longer rely on Uzawa-type algorithms, but there are still good reasons to reconsider them. They are simple, modular, and easy to implement in finite element software. This study revisits Uzawa methods with an emphasis on handling strong viscosity variations. Building on the work of Braess (Braess, 2001), who formulated the Schur complement as an explicit conjugate gradient problem, we show that two factors are especially important—the definition of residuals and the choice of preconditioner.

Our first extension changes how residuals are updated in the pressure space using inner products, which works the same as applying a mass-matrix preconditioner. This connects the  $L^2$ -projected residual update to preconditioned conjugate gradient (PCG) iterations and brings mass-matrix stabilization into the Uzawa framework (Elman and Golub, 1994; Ramage and Wathen, 1994; Chen, 1998; Benzi et al., 2005). We also introduce a viscosity-weighted variant where the mass matrix is replaced by an  $\eta$ -weighted version. This idea relates to earlier work on Schur complement preconditioners for variable-viscosity Stokes flow (May and Moresi, 2008). Unlike block preconditioning, our method puts viscosity weighting directly into the Uzawa iteration, making it a preconditioned conjugate gradient scheme with a preconditioner that better matches the true Schur complement spectrum. This gives faster and more stable convergence when viscosity contrasts are large.

The second extension is a post-processing projection step that projects velocity variable onto the constraint-satisfying subspace. For incompressible flow, this is the discrete Helmholtz-Hodge decomposition. Chorin (Chorin, 1967) first introduced this method for time-dependent Navier-Stokes equations, later refined for better stability and accuracy (Bell et al., 1989; Guermond and Quartapelle, 1998; Brown et al., 2001; Pyo and Shen, 2007). These methods are standard in unsteady fluid dynamics but have received little attention for elliptic Stokes problems in geodynamics. We use projection after Uzawa iterations to reduce constraint residuals.

The remainder of this paper is organized as follows. Section 2 presents the mathematical formulation, including the Stokes equations, the saddle-point system, Uzawa-type algorithms, residual representations, and the Helmholtz-Hodge projection.

Section 3 reports numerical experiments using manufactured benchmarks, incompressible and compressible mantle convection, and block sinking problems. Section 4 summarizes the main results and outlines future work.

#### 2 Mathematical formulation

# 60 2.1 Stokes equations and saddle-point formulation

We consider the incompressible Stokes equations relevant to mantle convection. Because the Prandtl number of mantle materials is effectively infinite ( $Pr \approx 10^{23}$ ), inertial terms in the Navier-Stokes equations can be neglected, and the governing system reduces to the elliptic Stokes equations (Schubert et al., 2001).

In a bounded domain  $\Omega \subset \mathbb{R}^{n_d}$   $(n_d = 2, 3)$ , these are

65 
$$\nabla \cdot \boldsymbol{\sigma} + \rho \mathbf{g} = \mathbf{0}, \quad \nabla \cdot \mathbf{u} = 0,$$

where **u** is velocity, p is pressure,  $\rho$  is density, and **g** is gravitational acceleration. The Cauchy stress tensor is

$$\sigma = -p\mathbf{I} + 2\eta D(\mathbf{u}), \qquad D(\mathbf{u}) = \frac{1}{2} (\nabla \mathbf{u} + (\nabla \mathbf{u})^T),$$

with  $\eta$  the dynamic viscosity and  $D(\mathbf{u})$  the strain-rate tensor.

In compact form, the Stokes equations then become

70 
$$\nabla \cdot (\eta(\nabla \mathbf{u} + (\nabla \mathbf{u})^T)) - \nabla p + \rho \mathbf{g} = \mathbf{0}, \quad \nabla \cdot \mathbf{u} = 0.$$
 (1)

In the case of constant viscosity, the momentum equation reduces to a vector Laplacian.

Discretization with stable finite element pairs (Babuška, 1973; Brezzi, 1974; Silvester and Wathen, 1994; Logg et al., 2012) gives the saddle-point system

$$\begin{pmatrix} K & G \\ G^T & 0 \end{pmatrix} \begin{pmatrix} u \\ p \end{pmatrix} = \begin{pmatrix} f \\ h \end{pmatrix}, \tag{2}$$

where K is the discrete viscosity matrix (symmetric positive definite) and G is the discrete gradient operator. The system is symmetric but indefinite, and efficient iterative solvers are required.

Eliminating velocity gives the Schur complement system

$$Sp = G^T K^{-1} f - h, \qquad S = G^T K^{-1} G.$$
 (3)

The Schur complement S is symmetric positive definite under the standard inf-sup stability condition. Forming S explicitly is too expensive, since each multiplication requires solving with  $K^{-1}$ , which is costly for large problems with variable viscosity. In practice, S is applied implicitly within iterative methods, together with suitable preconditioners.

85

# 2.2 Uzawa-type algorithms

For the Schur complement system (3), we consider three Uzawa-type methods: the standard Uzawa algorithm, its adaptive variant, and the conjugate-directions version by Braess (Braess, 2001). For convenience, we refer to the last method as the conjugate-directions Uzawa (CD-U).

**Standard Uzawa.** The standard Uzawa algorithm applies Richardson-type iterations to update the pressure and recompute the velocity. Given an initial pressure  $p^0$ , each iteration reads

$$Ku^{k+1} = f - Gp^k,$$

$$p^{k+1} = p^k + \omega (G^T u^{k+1} - h),$$
(4)

with relaxation parameter  $\omega > 0$ . The scheme is simple, but its convergence is sensitive to the choice of  $\omega$  and deteriorates with 90 large viscosity contrasts.

Adaptive Uzawa. To reduce this sensitivity, the relaxation parameter is updated adaptively at each step. With residual  $q^k = h - G^T u^k$  and auxiliary solve  $Kz^k = Gq^k$ , the update is

$$\omega_{k} = \frac{(q^{k})^{T} q^{k}}{(Gq^{k})^{T} z^{k}}, 
p^{k+1} = p^{k} - \omega_{k} q^{k}, 
u^{k+1} = u^{k} + \omega_{k} z^{k}.$$
(5)

This update formula improves robustness compared with the fixed- $\omega$  method.

Conjugate-directions Uzawa (CD-U). The CD-U method accelerates convergence by constructing conjugate directions, in analogy with the conjugate gradient method. Starting from  $p^0$  with

$$Ku^1=f-Gp^0, \qquad q^1=h-G^Tu^1, \qquad d^1=-q^1,$$

each iteration  $(k \ge 1)$  performs

$$Kz^{k} = Gd^{k}, \qquad \alpha_{k} = \frac{(q^{k})^{T}q^{k}}{(Gd^{k})^{T}z^{k}},$$

$$p^{k+1} = p^{k} + \alpha_{k}d^{k}, \qquad u^{k+1} = u^{k} - \alpha_{k}z^{k},$$

$$q^{k+1} = h - G^{T}u^{k+1}, \qquad \beta_{k} = \frac{(q^{k+1})^{T}q^{k+1}}{(q^{k})^{T}q^{k}},$$

$$d^{k+1} = -q^{k+1} + \beta_{k}d^{k}.$$
(6)

The explicit coefficients  $\alpha_k$  and  $\beta_k$  highlight the connection with CG. By exploiting information from previous steps, CD-U achieves faster and more stable convergence, especially for problems with strong viscosity contrasts or heterogeneous coefficients.

# 2.3 Residual representation in Uzawa iterations

The definition of the residual is central to Uzawa-type algorithms. In the Schur complement formulation, the discrete residual is

$$q^k = h - G^T u^k, (7)$$

which measures the violation of the incompressibility constraint. In its standard (vector) form,  $q^k$  is assembled directly as a discrete vector. This is simple but suffers from poor conditioning, and small singular values of  $G^T$  can lead to unstable step-size choices and slow convergence, especially with large viscosity contrasts.

An alternative is to define the residual variationally in the  $L^2$  inner product on the pressure space Q:

$$\langle q^k, r \rangle = \langle h + \nabla \cdot u^k, r \rangle, \quad \forall r \in Q.$$
 (8)

In discrete form this corresponds to solving a mass-matrix system,

$$M_{p}q^{k} = M_{p}h + b(u^{k}),$$

where  $b(u^k)$  denotes the discrete constraint term. This formulation shows that Uzawa iterations implicitly apply the pressure mass matrix as a preconditioner (Cahouet and Chabard, 1988; Elman and Golub, 1994; Chen, 1998; Benzi et al., 2005). The resulting  $L^2$ -projected residual is smoother and scale-independent, and yields more robust convergence at modest extra cost.

With this modification, the overall update structure of the standard, adaptive, and CD-U algorithms (cf. (4)-(6)) remains the same. Only the scalar coefficients change, as summarized below.

Standard Uzawa.

120 
$$\langle p^{k+1}, r \rangle = \langle p^k, r \rangle - \omega \langle q^{k+1}, r \rangle, \quad \forall r \in Q.$$
 (9)

**Adaptive Uzawa.** Using the  $L^2$  residual definition (8), the step size is updated as

$$\omega_{k+1} = \frac{\langle q^{k+1}, q^{k+1} \rangle}{\langle \nabla q^{k+1}, z^{k+1} \rangle}.$$
(10)

Conjugate-directions Uzawa (CD-U). With  $q^k$  defined by (8), the coefficients are

$$\alpha_k = \frac{\langle q^k, q^k \rangle}{\langle \nabla d^k, z^k \rangle}, \qquad \beta_k = \frac{\langle q^{k+1}, q^{k+1} \rangle}{\langle q^k, q^k \rangle}. \tag{11}$$

Conjugate-directions Uzawa with  $\eta$ -weighted residuals (CD-U- $\eta$ ). The standard  $L^2$  residual (8) can be extended by replacing the mass matrix with an  $\eta$ -weighted version. This introduces a viscosity scaling in the pressure space, which more closely reflects the spectrum of the true Schur complement for variable-viscosity problems. The resulting scheme is a preconditioned conjugate-gradient iteration in which both the residual  $q^k$  and its preconditioned form  $w^k$  appear in the update formulas.

130

140

Initialization: for all  $r \in Q$ ,

$$p^{0} = 0, Ku^{1} = f - Gp^{0},$$

$$\langle q^{1}, r \rangle = \langle h + \nabla \cdot u^{1}, r \rangle,$$

$$\langle \frac{1}{\eta} w^{1}, r \rangle = \langle q^{1}, r \rangle, d^{1} = -w^{1}.$$

$$(12)$$

Iteration  $(k \ge 1)$ : for all  $r \in Q$ ,

$$Kz^{k} = Gd^{k}, \qquad \alpha_{k} = \frac{\langle q^{k}, w^{k} \rangle}{\langle \nabla d^{k}, z^{k} \rangle},$$

$$p^{k+1} = p^{k} + \alpha_{k} d^{k}, \qquad u^{k+1} = u^{k} - \alpha_{k} z^{k},$$

$$\langle q^{k+1}, r \rangle = \langle h + \nabla \cdot u^{k+1}, r \rangle, \quad \langle \frac{1}{\eta} w^{k+1}, r \rangle = \langle q^{k+1}, r \rangle,$$

$$\beta_{k} = \frac{\langle q^{k+1}, w^{k+1} \rangle}{\langle q^{k}, w^{k} \rangle}, \qquad d^{k+1} = -w^{k+1} + \beta_{k} d^{k}.$$

$$(13)$$

This iteration can be interpreted as a Preconditioned Conjugate Gradient (PCG) method applied to the Schur complement system. Here  $q^k$  denotes the residual in the pressure space, and  $w^k$  its preconditioned form. The search directions  $d^k$  and coefficients  $\alpha_k, \beta_k$  are then defined from inner products such as  $\langle q^k, w^k \rangle$ , following the standard PCG structure.

With the  $L^2$  formulation, the preconditioner is the pressure mass matrix  $M_p$ , which is spectrally close to the identity. In this case  $w^k=q^k$ , and the scheme reduces to a Conjugate Gradient iteration with a trivial preconditioner. In the  $\eta$ -weighted formulation, the preconditioner is the viscosity-weighted mass matrix, so that  $w^k\neq q^k$  and both quantities appear in the iteration. Note that the relation between  $q^k$  and  $w^k$  cannot be expressed pointwise as  $w^k=\eta q^k$ , since  $\eta(x)$  varies spatially. Instead,  $w^k$  is defined variationally by

$$\langle \frac{1}{n} w^k, r \rangle = \langle q^k, r \rangle, \qquad \forall r \in Q,$$
 (14)

which ensures that  $w^k$  belongs to the pressure space Q and serves as the Riesz representative of  $q^k$  under the  $\eta$ -weighted inner product. This formulation provides a consistent SPD preconditioner in the finite element setting.

In summary, while the  $L^2$  formulation corresponds to a CG iteration with a trivial mass-matrix preconditioner, the  $\eta$ weighted Uzawa method is a genuine PCG scheme with a physically motivated preconditioner that better reflects the spectral
properties of the Schur complement, particularly under strong viscosity contrasts (May and Moresi, 2008).

#### 2.4 Helmholtz-Hodge projection as post-processing

Uzawa-type iterations reduce the constraint violation only asymptotically, so the discrete constraint remains slightly violated after a finite number of steps. To enforce the constraint explicitly, we apply a projection step as a post-processing. The idea is to correct an intermediate velocity  $\hat{u}$  by adding a gradient field so that the updated velocity satisfies the discrete constraint operator

$$C(u) = h$$
,

where  $C(u) = \nabla \cdot u$  for the incompressible case, and  $C(u) = \nabla \cdot (\bar{\rho}u)$  under the anelastic liquid approximation (ALA). For the incompressible case,  $C(u) = \nabla \cdot u$ , the projection requires solving a scalar Poisson problem

$$-\Delta \phi = h + \nabla \cdot \hat{u}, \qquad u^{k+1} = \hat{u} + \nabla \phi.$$

This corresponds to the discrete Helmholtz-Hodge decomposition, which states that any vector field can be uniquely decomposed into a solenoidal part and a gradient part,

$$\mathbf{u}^* = \mathbf{u} + \nabla \phi, \qquad \nabla \cdot \mathbf{u} = 0.$$

Equivalently, the projection operator can be written as

$$\mathcal{P} = I - G(G^T G)^{-1} G^T$$
,

which is symmetric and idempotent.

In variational form, the correction is obtained by solving

$$\langle \nabla \phi, \nabla \psi \rangle = \langle h + \nabla \cdot \hat{u}, \psi \rangle, \qquad \forall \psi \in Q,$$

and updating the velocity consistently in the finite element space,

$$\langle u^{k+1}, w \rangle = \langle \hat{u}, w \rangle + \langle \nabla \phi, w \rangle, \quad \forall w \in V.$$

This weak formulation is important: a direct discrete update  $u^{k+1} = \hat{u} + G\phi$  does not yield the same accuracy, whereas the variational form ensures consistency and better convergence. In practice, the projection can also be implemented componentwise to avoid assembling a large coupled system.

For the compressible case,  $C(u) = \nabla \cdot (\bar{\rho}u)$ , and the projection takes the weighted form

$$-\nabla \cdot (\bar{\rho}\nabla\phi) = h + \nabla \cdot (\bar{\rho}\hat{u}), \qquad u^{k+1} = \hat{u} + \nabla\phi,$$

so that  $\nabla \cdot (\bar{\rho}u^{k+1}) = h$  is satisfied exactly. This weighted projection is the natural extension of the Helmholtz-Hodge decomposition to density-dependent flows.

Although projection methods of this type are standard in computational fluid dynamics, they have not been used previously as post-processing corrections in geodynamics. As we show in Section 3, this step reduces accumulated divergence errors and improves the long-term accuracy of mantle convection benchmarks.

A related approach in particle-in-cell methods is the conservative velocity interpolation (CVI) scheme (Wang et al., 2015), which enforces local mass conservation during interpolation but follows a different formulation from the projection-based correction used here. The CVI approach is specifically designed for quadrilateral Q1P0 elements, whereas our projection method is element-independent and applicable to general finite element discretizations.

190

205

# 180 3 Numerical Implementation

#### 3.1 Finite Element Framework and Parallel Environment

We implemented the proposed algorithms using the open-source finite element library FEniCS (version 2019.2.0.13) (Logg et al., 2012; Alnæs et al., 2015). FEniCS provides a Python interface that simplifies variational form expression and automates finite element matrix assembly, which is useful in geodynamics where PDE systems often involve strongly variable rheologies (Zhamaletdinov et al., 2011; Vynnytska et al., 2013; Wilson et al., 2017; Lee et al., 2024). To solve the large sparse linear systems from discretizing the Stokes equations, we coupled FEniCS with PETSc (Balay et al., 2024). We used GMRES and CG with algebraic multigrid (AMG) preconditioning. Solver configurations were adjusted through PETSc runtime parameters.

All algorithms were implemented within the standard FEniCS-PETSc environment without custom code. This portability shows that the approach is reproducible, extendable, and transferable to other platforms, in line with current practices in large-scale geodynamic modeling.

# 3.2 Solver configuration and benchmark setup

Each Uzawa iteration with projection requires solving three linear systems: (i) the momentum equation for velocity, (ii) an  $L^2$ -projection in the pressure space for residual representation and scalar potential projection, and (iii) a Poisson system for velocity correction.

We initialized the pressure field to zero at the first step. In time-dependent simulations, we reused the pressure from the previous step as the initial guess, which improved convergence. The momentum equation was solved using GMRES with BoomerAMG from the Hypre library (Yang et al., 2002). The system matrix is symmetric positive definite, so CG could be used. However, in our tests GMRES proved more robust under strong viscosity contrasts and heterogeneous distributions. BoomerAMG builds algebraic multigrid hierarchies directly from the system matrix without explicit mesh information, and its parallel coarsening and smoothing strategies worked well for these systems.

The  $L^2$ -projection system was solved with CG preconditioned by block Jacobi, which works well for the diagonally dominant mass matrices from finite element discretizations. The Poisson system for velocity correction was also solved with CG and BoomerAMG. All solvers used relative and absolute tolerances of  $10^{-10}$  and  $10^{-12}$ . These tolerances are stricter than typical values in geodynamic simulations. We chose them to eliminate solver-dependent variations and ensure reproducible convergence across benchmarks. In practice, these values can be relaxed to  $10^{-6}$ – $10^{-7}$  without noticeably affecting accuracy.

Uzawa iterations were terminated using a normalized divergence residual

$$R_{\text{div}} = \frac{\|\nabla \cdot u^k\|_{L^2}}{\|u^k\|_{L^2}},$$

which measures how well the continuity constraint is satisfied. We used relaxed thresholds in long mantle convection runs to balance efficiency and accuracy.

All benchmarks were discretized with Taylor-Hood (P2-P1) finite elements. Geodynamic models often use quadrilateral or hexahedral meshes, but we used structured triangular and tetrahedral meshes generated by splitting each cell along its diagonals.

In two dimensions, this diagonal subdivision yields a symmetric "union-jack" pattern that preserves benchmark symmetry. This configuration provides robust velocity-pressure stability and straightforward implementation in FEniCS. Further discussion on finite element selection for geodynamic Stokes modeling can be found in (Thieulot and Bangerth, 2025). Viscosity  $\eta$  is represented as a finite-element field and evaluated at quadrature points during local assembly of element matrices.

To evaluate accuracy we report the relative  $L^2$  errors:

$$E_u = \frac{\|\mathbf{u} - \mathbf{u}_{\text{exact}}\|_{L^2}}{\|\mathbf{u}_{\text{exact}}\|_{L^2}}, \qquad E_p = \frac{\|p - p_{\text{exact}}\|_{L^2}}{\|p_{\text{exact}}\|_{L^2}}.$$

# 3.3 Baseline comparison of Uzawa variants

We first establish a baseline solver among the three Uzawa variants: standard Uzawa (S-U), adaptive Uzawa (A-U), and conjugate-directions Uzawa (CD-U). For this comparison, we use two complementary tests. The ABC flow with constant viscosity assesses convergence behavior in an ideal setting and is used to compare S-U, A-U, and CD-U directly. The SolCx problem with a viscosity contrast of  $10^6$  provides a stress test under strong heterogeneity; here we focus on CD-U and contrast three residual formulations—vector (algebraic),  $L^2$  inner-product, and the viscosity-weighted CD-U- $\eta$ —to examine robustness. In both tests we track the relative  $L^2$  errors in velocity and pressure and the normalized divergence residual over iterations, and select the most effective configuration for the remainder of the paper.

# 3.3.1 ABC Flow (smooth analytical solution)

The ABC (Arnold-Beltrami-Childress) flow (Zhao et al., 1993) is used as a smooth manufactured benchmark to compare the three Uzawa variants (S-U, A-U, CD-U) and to select a baseline solver. With constant viscosity ( $\eta=1.0$ ), all residual formulations reduce to the trivial case  $w^k=q^k$ . This setting isolates the algorithmic differences among the variants without interference from viscosity contrasts.

The exact velocity and pressure fields are

230

$$\mathbf{u}(x,y,z) = \begin{pmatrix} \sin(\pi z) + \cos(\pi y) \\ \sin(\pi x) + \cos(\pi z) \\ \sin(\pi y) + \cos(\pi x) \end{pmatrix}, \qquad p(x,y,z) = \sin(\pi x)\cos(\pi y)\cos(\pi z).$$

Figure 1 shows the relative velocity and pressure errors together with the normalized continuity residual  $R_{\rm div}$  as functions of iteration count. S-U converges very slowly and is highly sensitive to the relaxation parameter. A-U improves robustness but stagnates at higher error levels. CD-U, in contrast, converges rapidly and monotonically for all quantities. These results support choosing CD-U as the baseline solver for the following benchmarks.

Figure 1. ABC flow benchmark. Comparison of S-U, A-U, and CD-U: relative velocity error, pressure error, and normalized continuity residual  $R_{\text{div}}$  as functions of iteration count.

# 3.3.2 SolCx (viscosity contrast benchmark)

The SolCx problem (Zhong, 1996) is a standard two-dimensional benchmark for testing solver performance under strong viscosity contrasts. It is posed on  $\Omega = [0,1]^2$  with a discontinuous viscosity,

240 
$$\eta(x,y) = \begin{cases} 1, & x \le 0.5, \\ 10^6, & x > 0.5, \end{cases}$$

and a density field  $\rho(x,y) = \sin(\pi y)\cos(\pi x)$ . A gravitational acceleration of unit magnitude (|g| = 1) was applied in the vertical direction, and reference solutions were taken from Underworld (Mansour et al., 2020; Moresi et al., 2007).

This benchmark poses a challenging test for Uzawa-type solvers. Here we compare three variants of the Conjugate-Directions Uzawa method. The first is the discrete residual update (6), where the residual is updated in purely algebraic (vector) form. The second is the  $L^2$  inner-product formulation, in which the residual is defined variationally as in (8), and the update coefficients are computed using the inner-product relations in (11). Finally, we consider the viscosity-weighted extension CD-U- $\eta$ , initialized as in (12) and iterated according to (13), which realizes a preconditioned conjugate-gradient method on the Schur complement with a physically motivated  $\eta$ -weighted mass matrix.

Figure 2 shows that the discrete CD-U update converges slowly and stagnates at high pressure error. The  $L^2$  inner-product version achieves faster convergence but with oscillations in  $R_{\rm div}$ . The  $\eta$ -weighted variant provides the most rapid initial reduction of velocity and pressure errors and reaches much lower  $R_{\rm div}$  within a few iterations. At later steps,  $R_{\rm div}$  shows a mild increase. This occurs when conjugacy—the orthogonality of successive search directions in the preconditioned Schur complement inner product—degrades due to inexact inner solves for  $Kz^k = Gd^k$  and  $M_{\eta}w^k = q^k$ . Loss of conjugacy means that a new step partly points back toward earlier directions. Progress measured in the physical  $L^2$  divergence can be reversed locally, even though the preconditioned CG energy still decreases.

In our numerical experiments, we found that restarts or periodic reinitialization could alleviate this effect. We did not apply them here because the iteration counts are not large enough to require it. Subsequent benchmarks remain stable without

Figure 2. Convergence of CD-U variants on the SolCx benchmark. Relative velocity error  $E_u$ , pressure error  $E_p$ , and normalized divergence residual  $R_{\rm div}$  are compared for the discrete update, the  $L^2$  inner-product update, and the  $\eta$ -weighted update. All three share the same CD-U framework.

reinitialization. The small increase in  $R_{\rm div}$  is a characteristic of the baseline  $\eta$ -weighted formulation rather than a practical limitation.

# 260 3.4 Post-processing strategies for incompressible problems

We next examine the effect of the post-processing correction in two incompressible tests: linear mantle convection with variable viscosity and the block sinking problem with sharp density and viscosity contrasts. In both tests the mesh, boundary conditions, and time stepping are kept identical; the only change is in the Stokes solver. Results from CD-U- $\eta$  are compared with those from CD-U- $\eta$  combined with post-processing. Reference solutions are obtained with the direct LU solver MUMPS (Amestoy et al., 2000), which solves the discrete system to near machine precision.

Performance is evaluated by the relative  $L^2$  errors of velocity and scalar fields (temperature, composition, or density) and by the normalized continuity residual  $R_{\rm div}$ . A breakdown of solver cost is also reported, showing that the post-processing step adds little to the total runtime, while momentum solves remain the dominant cost.

## 3.4.1 Linear mantle convection

265

The linear mantle convection benchmark was solved on  $\Omega = [0,2] \times [0,1]$  using a  $320 \times 160$  structured grid, subdivided into triangular P2-P1 Taylor-Hood elements for velocity and pressure. Viscosity followed a depth- and temperature-dependent exponential law (Blankenbach et al., 1989),

$$\eta(x,y) = \exp\left(-b\frac{T}{\Delta T} + c(1-y)\right), \qquad b = \log(2.5), c = \log(2.0).$$

The body force in the Stokes equations was given by thermal and compositional buoyancy,

$$\mathbf{f} = (Ra_T T + Ra_{\phi} \phi) \hat{\mathbf{g}}, \qquad \hat{\mathbf{g}} = (0, -1),$$

Figure 3. Comparison of CD-U- $\eta$  and CD-U- $\eta$ +Post-processing over 5000 steps ( $\Delta t = 10^{-6}$ ). The post-processing step reduces errors in velocity, composition, and temperature fields, and maintains lower  $R_{\rm div}$  throughout the run.

with Rayleigh numbers

$$Ra_T = Ra_{\phi} = 10^6$$
.

Initial temperature was prescribed from boundary-layer theory (van Keken, 1997), and the composition field  $\phi$  was initialized as a dense basal layer (Tan and Gurnis, 2005). Boundary conditions were no-slip at the bottom and free-slip elsewhere for velocity; T=1 at the bottom and T=0 at the top with insulating sidewalls; and  $\phi=1$  at the bottom with homogeneous Neumann elsewhere.

The transport equations for temperature T and composition  $\phi$  are

$$\partial_t T + \mathbf{u} \cdot \nabla T - \nabla \cdot (\kappa \nabla T) = 0, \qquad \partial_t \phi + \mathbf{u} \cdot \nabla \phi = 0,$$

with  $\kappa=1$ . Both fields were discretized with P1 discontinuous Galerkin elements using upwind fluxes. Time integration employed the second-order BDF2 scheme, initialized by backward Euler. The Stokes system was solved at each step with CD-U- $\eta$  or CD-U- $\eta$  + Post-processing. Iterations were stopped once the normalized continuity residual  $R_{\rm div}$  fell below  $10^{-3}$ .

Figure 3 shows that the post-processing step improves accuracy in all fields. Velocity errors remain smaller at every stage, and the normalized continuity residual  $R_{\rm div}$  is consistently lower. Since the composition and temperature fields are advected by the velocity, they also benefit and show clear error reductions. Over 5000 steps, these corrections accumulate, yielding higher long-term accuracy compared with the baseline solver.

**Figure 4.** Reference fields (top) and absolute errors in composition and temperature at the final step (5000). Middle: CD-U- $\eta$ ; Bottom: CD-U- $\eta$ +Post-processing. The post-processing correction reduces localized interface errors and improves the accuracy of scalar advection.

Figure 4 shows composition and temperature errors at the final step. CD-U- $\eta$  alone produces noticeable errors near the interface, especially in the composition field. With post-processing, these localized errors are reduced and the solution is closer to the reference from the direct LU solver MUMPS, which is effectively exact for the discrete system. This demonstrates that post-processing improves accuracy and aligns the iterative solution more closely with that of a direct solver.

Table 1 summarizes the solver cost for CD-U and CD-U- $\eta$  with post-processing. The viscosity-weighted variant reduces the average number of momentum solves per step (2.40 versus 3.05) under the same tolerance, leading to an overall runtime reduction of about 10%. The extra mass-matrix operations in CD-U- $\eta$  (two solves per iteration instead of one) are inexpensive, contributing only about 1-2% of the total cost. Similarly, the post-processing corrections, consisting of a Poisson solve and velocity update, account for only a small fraction of the runtime compared with the dominant momentum solves. The main benefit of CD-U- $\eta$  is therefore the reduction in momentum iterations, while the additional scalar operations have negligible impact on efficiency.

**Table 1.** Cost breakdown for 5000 steps ( $\Delta t = 10^{-6}$ , tolerance  $10^{-3}$ , 32 cores). Only solver execution time is included. Both CD-U and CD-U- $\eta$  are shown with post-processing.

|                               | <b>CD-U</b> (Total 2036 s) |        |             | <b>CD-U-</b> η (Total 1849 s) |        |             |
|-------------------------------|----------------------------|--------|-------------|-------------------------------|--------|-------------|
| Step                          | Total Solves               | % Time | Solves/step | Total Solves                  | % Time | Solves/step |
| Momentum solves               | 15264                      | 73.2%  | 3.05        | 11998                         | 69.1%  | 2.40        |
| Mass-matrix solves (in Uzawa) | 15264                      | 0.5%   | 3.05        | 23996                         | 1.4%   | 4.80        |
| Post-processing (Poisson)     | 5000                       | 2.6%   | 1.00        | 5000                          | 3.0%   | 1.00        |
| Post-processing (velocity)    | 5000                       | 2.1%   | 1.00        | 5000                          | 2.3%   | 1.00        |
| Composition advection         | 5000                       | 3.3%   | 1.00        | 5000                          | 3.8%   | 1.00        |
| Temperature advection         | 5000                       | 18.3%  | 1.00        | 5000                          | 20.4%  | 1.00        |

## 3.4.2 Block sinking problem

The block sinking benchmark (Gerya, 2019) is a two-material Stokes flow test designed to evaluate solver robustness under sharp density and viscosity contrasts. A dense, low-viscosity block sinks into a lighter and more viscous fluid. The interface was represented by a DG0 level-set field ( $\phi$ ), positive inside the block, negative outside, and zero on the interface. The level-set was advected in time without reinitialization.

The domain  $\Omega=[0,1]^2$  was discretized with  $256\times 256$  triangular P2-P1 Taylor-Hood elements for velocity and pressure. The block initially occupied  $0.4 \le x \le 0.6, 0.7 \le y \le 0.9$ . Material properties showed strong contrasts: density 4200 vs. 2800, viscosity 0.1 vs. 100. All sides had free-slip boundaries, and the pressure was initialized uniformly. The body force was buoyancy,

$$\mathbf{f} = \rho(x, y) \,\hat{\mathbf{g}}, \qquad \hat{\mathbf{g}} = (0, -1),$$

with  $\rho$  defined by the block geometry. The level-set was advanced using a Crank-Nicolson scheme.

Solver iterations used a relaxed stopping condition  $R_{\rm div} 

Figure 5. Relative errors in velocity  $(E_u)$ , density  $(E_\rho)$ , level-set  $(E_\phi)$ , and  $R_{\rm div}$  over 1500 steps  $(\Delta t = 5 \times 10^{-6})$  for the block sinking problem.

**Table 2.** Solver cost breakdown for the block sinking simulation (1500 steps,  $\Delta t = 5 \times 10^{-6}$ , tolerance 0.2, 32 cores). Only solver execution time is included.

|                               | <b>CD-U</b> (Total 4168 s) |        |             | <b>CD-U-</b> η (Total 1735 s) |        |             |
|-------------------------------|----------------------------|--------|-------------|-------------------------------|--------|-------------|
| Step                          | Total Solves               | % Time | Solves/step | Total Solves                  | % Time | Solves/step |
| Momentum solves               | 33359                      | 98.5%  | 22.2        | 12523                         | 97.2%  | 8.35        |
| Mass-matrix solves (in Uzawa) | 33359                      | 0.8%   | 22.2        | 25046                         | 1.4%   | 16.7        |
| Post-processing (Poisson)     | 1500                       | 0.3%   | 1.00        | 1500                          | 0.7%   | 1.00        |
| Post-processing (velocity)    | 1500                       | 0.3%   | 1.00        | 1500                          | 0.8%   | 1.00        |
| Level-set advection           | 1500                       | 0.1%   | 1.00        | 1500                          | 0.2%   | 1.00        |

Table 2 compares the solver cost for CD-U and CD-U- $\eta$  with post-processing in the block sinking problem. The viscosity-weighted variant reduces the average number of momentum solves per step from 22.2 to 8.35, lowering the total runtime from 4168 s to 1735 s. The extra mass-matrix solves in CD-U- $\eta$  are inexpensive, contributing only 1-2% of the total. Post-processing corrections and level-set advection also remain minor. The main benefit of CD-U- $\eta$  is therefore the smaller iteration count in the momentum equations, while the additional scalar operations add little overhead.

**Figure 6.** Final-step density and level-set fields after 1500 steps ( $\Delta t = 5 \times 10^{-6}$ ). Left: reference solution (MUMPS). Middle: CD-U- $\eta$  only. Right: CD-U- $\eta$  with post-processing.

## 3.5 Compressible mantle convection

We next consider mantle convection under the Anelastic Liquid Approximation (ALA) (Schubert et al., 2001; King et al., 2008), which incorporates depth-dependent density while filtering out fast acoustic modes. This benchmark follows (King et al., 2008) with Rayleigh number  $Ra = 10^5$  and dissipation number Di = 0.5. This case extends the projection-based correction to compressible formulations, where mass conservation is weighted by the background density profile.

Under ALA, mass conservation is expressed as

$$\nabla \cdot (\bar{\rho} \mathbf{u}) = 0, \qquad \bar{\rho}(z) = \rho_0 \exp\left(\frac{Di(1-z)}{\gamma_0}\right).$$

335 The momentum equation is

$$\nabla \cdot \boldsymbol{\tau}(\mathbf{u}) - \nabla p' + Ra \,\bar{\rho} \,\alpha \,T \,\hat{\mathbf{k}} - \left(\frac{Di}{\Gamma} \frac{c_p}{c_v}\right) \bar{\rho} \,\chi_T \,p' \,\hat{\mathbf{k}} = \mathbf{0},$$

where

$$\boldsymbol{\tau}(\mathbf{u}) = 2\mu\,\boldsymbol{\varepsilon}(\mathbf{u}) - \tfrac{2}{3}\mu(\boldsymbol{\nabla}\cdot\mathbf{u})\,\boldsymbol{I}, \qquad \boldsymbol{\varepsilon}(\mathbf{u}) = \tfrac{1}{2}\big(\boldsymbol{\nabla}\mathbf{u} + \boldsymbol{\nabla}\mathbf{u}^\top\big).$$

The temperature equation, including advection, diffusion, adiabatic heating, and viscous dissipation, is

340 
$$\bar{\rho}c_p\,\partial_t T + \bar{\rho}c_p\,\mathbf{u}\cdot\nabla T - \nabla\cdot\left(\kappa\nabla(\bar{T}+T)\right) + \alpha\,\bar{\rho}\,Di\,u_y\,T - \frac{Di}{Ra}\,\boldsymbol{\sigma}(\mathbf{u}):\nabla\mathbf{u} = 0,$$

345

with background profile

$$\bar{T}(z) = T_0 \exp(Di(1-z)).$$

Time discretization used a second-order Crank-Nicolson scheme. Since viscosity is constant, CD-U and CD-U- $\eta$  coincide; the focus here is therefore on testing the robustness of the projection-based correction under compressible mass conservation.

To enforce the weighted continuity constraint after each Uzawa iteration, a  $\bar{\rho}$ -weighted velocity correction was applied. Given an intermediate velocity  $\hat{\mathbf{u}}$ , we solved

$$-\nabla \cdot (\bar{\rho}\nabla\phi) = \nabla \cdot (\bar{\rho}\hat{\mathbf{u}}), \qquad \mathbf{u} = \hat{\mathbf{u}} + \nabla\phi,$$

ensuring  $\nabla \cdot (\bar{\rho} \mathbf{u}) = 0$  at the discrete level.

The computational domain  $\Omega=[0,1]^2$  was discretized into  $160\times160$  triangular Taylor-Hood (P2-P1) elements for velocity and pressure, and P1 discontinuous Galerkin elements for temperature. Viscosity was constant ( $\eta=1$ ). Free-slip boundaries were imposed on the horizontal sides and no-slip on the vertical sides. Temperature was fixed at the bottom ( $T\approx0.09$ ) and top (T=0), with insulated sidewalls. Initial conditions included sinusoidal perturbations of the background temperature, and pressure was initialized to zero. The Uzawa method with conjugate directions (CD-U) was used as the baseline solver, with convergence tolerance  $R_{\rm div}^{\bar{p}}

Figure 7. Relative  $L_2$  errors in velocity  $(E_u)$  and temperature  $(E_t)$  over 5000 steps  $(\Delta t = 10^{-6})$  for compressible mantle convection under ALA. The  $\bar{\rho}$ -weighted velocity correction improves accuracy compared with CD-U alone.

Figure 7 shows that with constant viscosity the CD-U solver alone is already robust, but the  $\bar{\rho}$ -weighted correction further reduces errors in velocity and temperature. Figure 8 confirms improved accuracy near steep thermal gradients.

In terms of cost, the constant-viscosity ALA setup required only about 2-3 momentum solves per step on average, making it considerably cheaper than variable-viscosity cases. The  $\bar{\rho}$ -weighted correction added less than 20% to the runtime, confirming that its accuracy benefits come at modest overhead.

**Figure 8.** Reference temperature field (left, MUMPS) and absolute temperature errors at the final step (5000). Middle: CD-U only; Right: CD-U with  $\bar{\rho}$ -weighted correction. The correction reduces localized errors near steep gradients.

## 4 Conclusions

365

375

380

We revisited Uzawa-type algorithms for Stokes equations in geodynamics and propose two enhancements. The residual calculation was reformulated in variational form, which provides a natural preconditioner. We then extended this to a viscosity weighting version (CD-U- $\eta$ ). This can be interpreted as a preconditioned conjugate gradient method for the Schur complement. The second enhancement applies a Helmholtz-Hodge projection as post-processing to the velocity field. This enforces the constraint more accurately. Through several benchmark tests, we confirmed that these two approaches help the algorithm obtain more accurate solutions.

Our benchmarks include smooth analytical flows, high-contrast SolCx, linear mantle convection, block sinking with multiphase interfaces, and compressible mantle convection under ALA. CD-U- $\eta$  converged faster and stayed robust under large viscosity contrasts. The projection step improved constraint satisfaction and enhanced long-term accuracy. In time-dependent simulations, small corrections at each step accumulated and substantially improved transported fields such as temperature, composition, and density.

In our benchmark tests, the additional cost of  $L^2$  residual and projection corrections was modest—less than 5% of total solver time. Momentum solves remained the dominant component. The enhancements do not significantly change overall cost. As problem size increases, the relative cost of these corrections should decrease further since momentum solves become more dominant. Additionally, the viscosity-weighted variant helps reduce momentum iterations, further improving computational efficiency.

It is important to note that the projection step only corrects the velocity field and does not directly enhance pressure convergence. This highlights an inherent limitation of projection-based strategies, which are designed to enforce incompressibility. Since pressure plays a critical role in many geodynamic applications, its convergence remains governed by the underlying Uzawa iteration. Our focus here was to improve velocity and advected fields. Improving pressure convergence represents an important direction for future research.

https://doi.org/10.5194/egusphere-2025-5480 Preprint. Discussion started: 18 November 2025

© Author(s) 2025. CC BY 4.0 License.

385

EGUsphere Preprint repository

In conclusion, projection-enhanced Uzawa solvers offer a practical alternative to more complex Schur complement methods. They are easy to implement in standard FEM frameworks like FEniCS-PETSc, work across different platforms, and perform well for both incompressible and compressible mantle convection problems with strong rheological complexity. These findings suggest that projection-enhanced Uzawa solvers can serve as a useful baseline for future work in geodynamic modeling, combining simplicity with computational efficiency for large-scale problems.

Code availability. The numerical benchmarks used in this study (ABC flow, SolCx, mantle convection, block sinking, and compressible convection) are publicly available and documented in the cited references. All simulations were performed using the open-source finite element library FEniCS (version 2019.2.0.13) coupled with PETSc. The implementation code will be made available upon acceptance at a Zenodo repository. Specific solver configurations and benchmark parameters are described in Sect. 3.

Author contributions. D.-K. J. designed the algorithms, implemented the software, and prepared the first draft of the manuscript.

- K.-M. L. carried out the numerical experiments, performed the data analysis and contributed to the review of the manuscript.
- C. T. contributed substantially to shaping the core ideas of the paper through revisions and provided major input during the manuscript editing process.
  - W.-H. C. assisted in numerical implementation, provided feedback on data interpretation, and contributed to securing financial support.
  - B.-D. S. supervised the project, acquired funding, and revised the manuscript.

All authors discussed the results and contributed to the final version of the paper.

Competing interests. The authors declare that they have no conflict of interest.

Acknowledgements. This work was supported by the National Research Foundation of Korea (NRF) grants funded by the Korea government (No. RS-2025-00519120 to Deok-Kyu Jang, Nos. RS-2025-25415913 and NRF-2022R1C2011689 to Whan-Hyuk Choi, and Nos. RS-2025-02293161 and NRF-2022R1A2C1009742 to Byung-Dal So).

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
