# Peer review of "Efficient Uzawa algorithms with projection strategies for geodynamic Stokes flow"

_EGUsphere, 2025_

## Referee Comment (RC1)

**Report for: Efficient Uzawa algorithms with projection strategies for gedynamical Stokes flow**

by Jang, Lee, Thieulot, Choi and So

**Summary**

The manuscript introduces Uzawa-type solvers for the linear Stokes equations with strong viscosity variations. It extends existing approaches in several directions, including the use of (inverse) viscosity-weighted mass matrices in the pressure Schur complement and the application of a velocity projection to better enforce incompressibility. The paper describes and contrasts these methods, but does not include any convergence analysis. Instead, through a series of interesting numerical experiments, the authors study the methods' performance in terms of accuracy and runtime.

As the authors acknowledge in the introduction, Uzawa methods are no longer regarded as state-of-the-art for Stokes problems; contemporary approaches typically handle velocity and pressure simultaneously and generally achieve superior convergence. The authors argue that Uzawa methods are simple, modular, and therefore easy to implement. I am not entirely persuaded by this claim. Given the solver technology available in modern FEM and linear algebra libraries (such as FEniCS and PETSc), implementing a full-space Stokes solver with a block-diagonal preconditioner is not particularly difficult.

Overall, this is a carefully prepared and clearly written manuscript that I enjoyed reading. I did not identify any major issues with the exposition or the writing style. My primary reservation, however, concerns the broader question raised above: at what point should we stop using and further developing methods that are outperformed by current state-of-the-art approaches that are already available in open-source libraries (for example, Aspect)? I leave the final decision to the editor. If he/she concludes that the manuscript contains sufficient and relevant novelty to merit publication, I would ask the authors to consider incorporating some of the suggestions I outline below.

**Issues/suggestions:**

- l71 – As for the statement that the constant-viscosity Stokes system reduces to the vector Laplacian: disregarding boundary conditions this is certainly true, and it also holds for Dirichlet conditions. But is it still generally valid for more complex boundary conditions?

- Throughout the paper, the authors refer to the $\eta$-weighted version of the preconditioner. Is this weight not actually the inverse viscosity?

- l148: The authors argue that the divergence-free condition is slightly violated due to inexact solves and that this is corrected by the Helmholtz projection. I can see this being the case if Taylor–Hood elements are also used in the projection step, which appears to be what is done. Would this issue vanish entirely if more iterations were used so that the system is solved more accurately? This could be investigated further in the numerical experiments.

- Figure 1 (and, in fact, all experiments) use a *fixed* discretization. Yet, one of the most interesting and important properties of numerical methods is their behavior under mesh refinement. It would be valuable to show results for at least one level of finer mesh.

- I would conjecture that the benchmarks in Sec. 3.3.1 and 3.3.2 have rather smooth solutions, so I am not convinced that these are the most informative problems to consider. The May/Moresi sinker example (Sec. 3.4.2) is more demanding, although a multi-sinker setup (e.g., May et al., SC Proceedings, 2014; Rudi et al., SISC 2017) is known to be even more challenging and would definitely be insightful.